# Association of *UBE3C* Variants with Reduced Kidney Function in Patients with Diabetic Kidney Disease

**DOI:** 10.3390/jpm10040210

**Published:** 2020-11-06

**Authors:** Ying-Chun Chen, Mei-Yi Wu, Zhi-Lei Yu, Wan-Hsuan Chou, Yi-Ting Lai, Chih-Chin Kao, Imaniar Noor Faridah, Mai-Szu Wu, Wei-Chiao Chang

**Affiliations:** 1Master Program in Clinical Pharmacogenomics and Pharmacoproteomics, School of Pharmacy, Taipei Medical University, Taipei 11031, Taiwan; 12556@s.tmu.edu.tw (Y.-C.C.); e220121@gmail.com (M.-Y.W.); 2Department of Pharmacy, Shuang Ho Hospital, Taipei Medical University, New Taipei City 23561, Taiwan; 3Division of Nephrology, Department of Internal Medicine, School of Medicine, College of Medicine, Taipei Medical University, Taipei 11031, Taiwan; salmonkao@gmail.com; 4Division of Nephrology, Department of Internal Medicine, Shuang Ho Hospital, Taipei Medical University, New Taipei City 23561, Taiwan; 5TMU Research Center of Urology and Kidney, Taipei Medical University, Taipei 11031, Taiwan; 6Department of Clinical Pharmacy, School of Pharmacy, Taipei Medical University, Taipei 11031, Taiwan; r06a41003@ntu.edu.tw (Z.-L.Y.); s700081@gmail.com (W.-H.C.); s8901752004@gmail.com (Y.-T.L.); imaniar.faridah@pharm.uad.ac.id (I.N.F.); 7Faculty of Pharmacy, University of Ahmad Dahlan, Yogyakarta 55164, Indonesia; 8Integrative Research Center for Critical Care, Wan Fang Hospital, Taipei Medical University, Taipei 11696, Taiwan; 9Department of Medical Research, Shuang Ho Hospital, Taipei Medical University, New Taipei City 23561, Taiwan

**Keywords:** diabetic kidney disease, UBE3C, genetic polymorphism

## Abstract

Diabetic kidney disease (DKD) is the leading cause of morbidity and mortality in patients with diabetes mellitus (DM) and the most common variant of end-stage renal disease (ESRD) globally. The economic burden of ESRD treatment with dialysis is substantial. The incidence and prevalence of ESRD in Taiwan remain the highest worldwide. Therefore, identifying genetic factors affecting kidney function would have valuable clinical implications. We performed microarray experiments and identified that ubiquitin protein ligase E3C (*UBE3C*) is differentially expressed in two DKD patient groups with extreme (low and high) urine protein-to-creatinine ratios. A follow-up genotyping study was performed in a larger group to investigate any specific variants of *UBE3C* associated with DKD. A total of 263 patients were included in the study, comprising 172 patients with DKD and 91 control subjects (patients with DM without chronic kidney disease (CKD)). Two *UBE3C* variants (rs3802129(AA) and rs7807(CC)) were determined to be associated with reduced kidney function. The haplotype analysis revealed that rs3802129/rs3815217 (block 1) with A/G haplotype and rs8101/rs7807 (block 2) with T/C haplotype were associated with higher risks of CKD phenotypes. These findings suggest a clinical role of *UBE3C* variants in DKD risk.

## 1. Introduction

Independent of demographic characteristics and hypertension, diabetes mellitus (DM) is strongly associated with albuminuria and a reduced glomerular filtration rate (GFR) [1]. Microvascular changes in the kidney often lead to chronic kidney disease (CKD), referred to as diabetic kidney disease (DKD) or diabetic nephropathy. The increased prevalence of DKD is the leading cause of high prevalence and incidence of end-stage renal disease (ESRD) in Taiwan and developed countries [2,3]. The standard management for kidney disease focuses on treating the underlying conditions, lifestyle modification, and dietary control. Specific anti-glycemic (e.g., glucagon-like peptide-1 receptor agonists and sodium glucose cotransporter 2 inhibitors) and antihypertensive agents (e.g., angiotensin-converting enzyme inhibitors and angiotensin II receptor antagonist) have reno-protective effects and slow the disease progression [4,5,6,7,8]. However, intensive glucose control to achieve stringent targets could also negatively affect mortality [9,10]. Furthermore, some patients experience a relatively rapid deterioration of renal function, whereas others maintain normal renal function despite decades of suboptimal glycemic control [11].

Reports from DKD in the families of individuals with type I DM suggest a critical role of genetic factor [12,13]. At the transcriptomic level, previous studies identified several differentially expressed genes in glomeruli and tubuli of DKD samples, including the chemokines, adhesion molecules and inflammatory cytokines [14,15]. The activation of NF-κB and AP-1 were found. In addition, an increased expression of TGF-β1 were observed in peripheral mononuclear cells of diabetic patients with nephropathy [16]. Multiple DKD susceptibility genes have been identified from biological candidate gene association studies and hypothesis-free genome-wide association studies (GWAS). The genes identified included SLC12A3 (a thiazide-sensitive Na–Cl cotransporter) [17], ELMO1 (the engulfment and cell motility 1 gene) [18], and genetic locus on human chromosome 6q25.2 (rs955333) between the SCAF8 and CNKSR genes [19]. Results from GWAS revealed a novel signal near GABRR1 associated with “microalbuminuria” in European patients with type 2 DM [20]. However, few of these susceptibility loci were robustly and consistently replicated in studies of transethnic populations. Here, we used microarray screening to identify ubiquitin protein ligase E3C (*UBE3C*) as a critical gene that associated with kidney functions in Taiwanese patients with DM. *UBE3C* was significantly associated with end-stage kidney disease, which accorded with a prior study [21]. We further identified e novel *UBE3C* gene variants and haplotypes that significantly associated with kidney disease among patients with DM.

## 2. Materials and Methods

### 2.1. Study Subjects

A schematic of the study design is displayed in Figure 1. The study participants were recruited from Taipei Medical University Shuang Ho Hospital (SHH) and Taipei Medical University Hospital (TMUH). Patients were eligible if they were aged 18 years or older and were diagnosed as having type 2 DM, defined based on diagnosis criteria by the American Diabetes Association, international classification of diseases, 9th revision, clinical modification (ICD-9-CM) code 250, ICD-10-CM code E11, or SHH or TMUH electronic medical records on prescriptions. Exclusion criteria were a history of any of the following: type 1 DM, gestational diabetes, maturity-onset diabetes, and kidney transplantation. A total of 263 participants were recruited. The study protocol was approved by the Taipei Medical University Joint Institutional Review Board (TMU-JIRB N201411056, and TMU-JIRB N201309026).

The age, sex, body mass index (BMI), HbA1c, serum creatinine, urine creatinine, urine protein, and comorbidity status of patients were recorded for further analyses. Estimated GFR (eGFR) was calculated using the four-variable modification of diet in renal disease equation, and urine protein-to-creatinine ratio (UPCR) was assessed by performing a spot measurement of urine protein and creatinine.

Two binary phenotypes were defined to explore the disease severity spectrum and the different disease processes represented by eGFR or proteinuria. First, the CKD phenotypes in a case group with an eGFR of <60 mL/min/1.73 m^2^ and a control group with an eGFR of ≥60 mL/min/1.73 m^2^ were compared to identify variants that contribute to reduced kidney function. Second, the proteinuria phenotypes of the case group with a UPCR of >150 mg/g and a control group with a UPCR of ≤150 mg/g were compared to detect variants that contribute to severe glomerular barrier dysfunction. The phenotype definitions were aligned with other large-scale genetic studies of type 2 DM DKD cohorts [17]. The definition of CKD was also aligned to the definition used by the CKDGen Consortium (eGFR <60 mL/min/1.73 m^2^) [22], although case and control groups were restricted to patients with diabetes.

### 2.2. Microarray Experiment

Microarrays were performed for eight patients with extreme UPCR levels to compare their transcriptomics profiles. RNA was extracted from peripheral blood mononuclear cells using the RNeasy Plus Mini Kit (QIAGEN, Inc., Hilden, Germany). The quality of the samples was assessed using a Qsep100 DNA Analyzer, NanoDrop 2000 Spectrophotometer, and 2100 Bioanalyzer (Agilent Technologies, Inc., CA, USA). Microarray experiments were performed at the High-Throughput Genome and Big Data Analysis Core Facility of the National Yang-Ming University VGH Genome Research Center (VYMGC) using the Affymetrix Human U133 plus 2.0 gene chip.

### 2.3. Criteria for the Selection of UBE3C Variants

The location of *UBE3C* was identified using the UCSC Genome Browser (http://genome.ucsc.edu/) on Human Feb. 2009 (GRCh37/hg19) assembly version. The Han Chinese in Beijing (CHB) population genome data were downloaded from 1000 Genomes Browser Ensembl GRCh37 release 92, April 2018 version. The tag single-nucleotide polymorphisms (tSNPs) were selected using Haploview 4.2 program if they had an r^2^ of 0.8 or higher; the *p* value cutoff was above 0.001 at the Hardy–Weinberg equilibrium. The minimum minor allele frequency of the selected tSNPs was determined to be above 0.1.

### 2.4. Genotyping and eQTL Analysis

DNA was extracted from the whole-blood samples of patients with DM using the Gentra Puregene Blood Kit (QIAGEN, Inc., Hilden, Germany). TaqMan SNP genotyping assays were used to determine the genotypes. A polymerase chain reaction (PCR) was performed using the StepOnePlus real-time PCR system. The thermal cycling started at 95 °C for 10 min, followed by 45 cycles of denaturing at 92 °C for 15 s. Finally, annealing and extension were conducted at 60 °C for 1 min. Allelic discrimination was then analyzed using StepOne software version 2.22 (Applied Biosystems part of Life Technologies Corp.). To further understand the functional effects of variants, eQTL analysis was performed. Data sets for the expression quantitative trait locus (eQTL) analysis were based on publicly available eQTL association datasets from the Genotype–Tissue Expression Project (GTEx)V7.

### 2.5. Statistical Methods and Analytical Tools

R version 3.4.3 was used for statistical analysis. The microarray data were normalized using affyPLM packages and the robust multi-array average method, which is a regular normalization approach. Differentially expressed genes (DEGs) in paired sample data were identified using moderated paired t tests provided by limma. The biomaRt package was used for probe-to-gene annotation. ComplexHeatmap and R package dendextend were used to perform unsupervised, agglomerative hierarchical clustering to visualize the data as a heatmap. The R package ggplot2 was used to visualize most of the data. Gene differences with unadjusted *p* values of less than 0.01 and absolute ln values (fold change) of more than 0.5 were considered significant DEGs. Statistical differences in categorical variables (sex and smoking status) were determined using chi-squared (χ2) tests. A Shapiro–Wilk test was used to determine the normality of each continuous variable. Nonparametric methods (Mann–Whitney) were used to compare the means of nonnormally distributed continuous variables (age, BMI, and HbA1c) between the patient groups. The association between haplotypes and the presence of CKD in patients with DM were further investigated. The Haploview 4.2 program was used to determine haplotypes. Standardized coefficients of linkage disequilibrium (LD) between neighboring single-nucleotide polymorphism (SNPs) (D’ value) above 0.95 were considered haplotype blocks in the LD. Haplotype association analysis was performed using the haplo.stat package with a case–control design. Furthermore, a multiple logistic regression model adjusted for potential confounders (age and sex) was employed for genotyping analysis. The associations between genotypes and risks of DKD were tested in the recessive model using the SNPassoc package [23]. Odds ratios (ORs) and 95% confidence intervals (CIs) were estimated. The false discovery rate was applied and *q* values were estimated to control for type I errors as a result of multiple testing [24].

## 3. Results

### 3.1. Demographic and Clinical Characteristics of the Study Sample

The demographic characteristics of study subjects for microarray screen are presented in Appendix A. Eight patients were recruited with age and gender matched. In addition, patients with extreme expression levels of UPCR (high and low UPCR) were matched with known risk factors, including BMI, GFR, HbA1c, and smoking history. All of the patients were male, and the high and low UPCR groups had mean ages of 67 ± 10.55 and 65.25 + 15.92 years, respectively. The mean UPCR levels were 7048.90 ± 2900.06 mg/g in the high UPCR group and 646.83 ± 227.25 mg/g in the low UPCR group.

In the experimental design for genetic association study, 263 patients were recruited (172 patients with DM and an eGFR of less than 60 mL/min/1.73 m^2^ and 91 subjects with DM and an eGFR of more than 60 mL/min/1.73 m^2^). The mean eGFR in the case group was 32.81 mL/min/1.73m^2^, and the mean eGFR in the control group was 89.58 mL/min/1.73m^2^. UPCR, another critical indicator of kidney failure, was 2174.52 ± 3090.89 and 784.51 ± 1461.02 mg/g in the case and control groups, respectively. The demographic data of study population was shown as Table 1. 

### 3.2. Microarray-Based Candidate Genes Selection from Eight Patients

Eight DM patients have been divided as high UPCR group and low UPCR group. The differentially expressed genes (DEG) analysis revealed 20 down-regulated DEGs and 11 up-regulated DEGs. Significant DEGs were clustered to clarify the expression pattern in patients with high and low expression levels of UPCR. Three clusters with opposite patterns were identified in the heat map presented in Figure 2. A total of 10 DEGs with opposite patterns were identified when the high UPCR group was compared with the low UPCR group (Appendix A). Among the DEGs, *UBE3C* displayed the most significant pattern between two groups.

### 3.3. Association of rs3802129 and rs7807 with Reduced Kidney Function

To further understand the clinical role of *UBE3C* variants, six tag single-nucleotide polymorphisms (tSNPs) on human *UBE3C* gene were selected for genotyping. Two SNPs (rs8101 and rs7807) are located in the 3′-prime-untranslated region (3′-UTR), whereas the other four (rs3802129, rs3815217, rs6979947, and rs12669987) are located in the intronic region. Two SNPs (rs75371354 (chr7:157047611) and rs138418294 (chr7:156993835)) were excluded because they failed the quality control of SNP probe design and no substitute SNP was available. Figure 3 presents a graphic overview of genotyped *UBE3C*. The position, location, alteration, and allele frequencies in different populations of the six tSNPs are shown in Table 2. The frequency of each SNPs between the current study and Taiwan biobank is similar, indicating appropriate quality of genotyping. In addition, frequencies of rs3815217 and rs12669987 were higher in Taiwanese and Asian populations compared to other populations. The frequencies of rs3802129 and rs6979947 were low in African populations. The frequencies of rs8101 were high in African populations, and the frequencies of rs7807 were high in European populations.

The comparison of patients with and without CKD revealed that four *UBE3C* variants (rs3802129, rs3815217, rs8101, and rs7807) were significantly associated with CKD in the recessive model after adjustment for age and sex (*p* values = 0.008, 0.032, 0.045, and 0.003, respectively). Two variants (rs3802129 and rs7807) were significantly associated with reduced kidney function (eGFR < 60 mL/min/1.73 m^2^) after adjustment for multiple testing (*q* value = 0.024 and 0.018, respectively). The patients with DM and rs3802129 AA genotype had higher risks of CKD than those with GG or AG genotypes (OR = 2.66; 95% CI = 1.24–5.68). Patients with the rs7807 CC genotype were more likely to have reduced kidney function compared with AA or AC genotype (OR = 3.76; 95% CI = 1.39–10.13). However, the other two variants (rs3815217 and rs8101) did not exhibit significant differences after correction for multiple testing (*q* value = 0.064 and 0.068, respectively) (Table 3). A comparison of patients with and without proteinuria did not reveal any significant associations between polymorphisms of *UBE3C* and UPCR (Appendix A).

### 3.4. Haplotype Association Analysis and Expression Quantitative Trait Locus Analysis

We detected two regions of strong LD blocks in our study cohort (Figure 4). The first block was composed of rs3802129/rs3815217, and the second block was composed of rs8101/rs7807 on the 3′-UTR of *UBE3C*. The rs3802129/rs3815217 (block 1) pairwise allele analysis revealed that patients with the A/G haplotype had a significantly higher risk of CKD phenotype compared with G/A haplotype (*p* value = 0.005; OR = 1.76; 95% CI = 1.19–2.61). In the haplotype block 2 (rs8101/rs7807), the T/C haplotype was significantly associated with a higher risk of CKD phenotype compared with the C/A haplotype (*p* value = 0.024; OR = 1.58; 95% CI = 1.06–2.35) (Table 4). To further confirm the biological functions of each SNPs, eQTLs analysis was performed. As shown in Table 5, among the DKD-associated variants, rs3802129, rs8101, and rs7807 were significant eQTLs in *UBE3C* according to GTEx. Furthermore, patients carrying risk alleles (rs3802129(A), rs8101(T), and rs7807(C)) displayed higher expressions levels of *UBE3C* than other alleles.

## 4. Discussion

By microarray screening, we determined that *UBE3C* is differentially expressed in DKD patients with extreme high and low UPCR levels. Furthermore, we identified two *UBE3C* variants (rs3802129(AA) and rs7807(CC)) that associated with reduced kidney functions. Haplotype analysis further confirmed the roles of *UBE3C* haplotype in the risks of CKD phenotypes. Proteinuria and eGFR are valuable biomarkers of dysfunction of the glomerular barrier and reduced kidney function, respectively, which have been used for the clinical diagnosis of DKD. However, these two features can develop independently among patients with DKD. Therefore, we performed two phenotypic comparisons to investigate *UBE3C* genetic predisposition. We obtained positive findings for eGFR-based phenotypes (Table 3) and negative findings for UPCR-based phenotypes (Appendix A), which indicates that the two main features of DKD may have different genetic origins and may involve distinct disease mechanisms. Our results are in line with a prior GWAS that identified distinct genetic variants in albuminuria-based phenotypes and eGFR-based phenotypes [20]. The DKD-related variants rs3802129, rs8101, and rs7807 were significantly correlated with *UBE3C* expression in the esophagus or skeletal muscle, determined using GTEx. Furthermore, previous studies have indicated that these variants are eQTLs of *UBE3C* in peripheral blood cells [25,26,27].

The *UBE3C* gene is ubiquitously expressed across most tissues and is overexpressed in the testis and esophagus. *UBE3C* encodes ubiquitin-protein ligase E3C, which plays an essential role in the ubiquitin-mediated degradation pathway. Conjugation of ubiquitin to the protein substrate occurs through a three-step sequential reaction [28]. The ubiquitin-activating enzyme (E1) activates ubiquitin, whereas ubiquitin-conjugating enzymes (E2) transfer the activated ubiquitin moiety from E1 to the substrate specifically bound to a member of the ubiquitin-protein ligase family E3 (UBE3C) [29]. The ubiquitin–proteasome system (UPS) selectively degrades misfolded proteins and intersects with the endoplasmic-reticulum-associated degradation (ERAD) pathway in the pathogenesis of proteinuric kidney disease [30].

A previous study nicely specified the importance of UPS in the regulation for physiological and pathophysiological processes along the entire length of the nephron. Mutations in E3 ligases complex (e.g., KLH3 and CUL3 or VHL) lead to altered protein degradation by the UPS and subsequently to kidney disease [31]. The UPS is upregulated in DKD [32], and high glucose stimulus can enhance proteasomal activity [33]. The inhibition of UPS may prevent oxidative stress, fibrosis, and inflammation in DKD by activating the Nrf2-Keap1 signaling pathway [34]. Furthermore, UBE3C has been reported in the regulation of multiple inflammatory diseases. For example, *UBE3C* gene expression was increased in renal cell carcinoma tissues compared with adjacent normal tissues [35]. An exome analysis revealed frequent mutations of the *UBE3C* gene in human hepatocellular carcinoma [36]. Polymorphisms in *UBE3C* were identified as potent markers of nasal polyps and aspirin-intolerant asthma among Korean patients with asthma [37,38]. Additionally, a large-scale GWAS identified variants in *UBE3C* that are associated with a high risk of type 2 DM in European populations [39]. Thus, the role of E3 ligase has been highlighted in the pathogenesis and progression of various diseases, such as obesity, diabetes, cancer, neurodegenerative disorders, and inflammatory diseases [40,41]. Targeting UPS and its components (including E3 ligases) may be valuable for identifying novel therapeutic targets for DKD.

However, this study has certain limitations. First, crucial information, such as the duration of DM, was unavailable in our data set. In the current analyses, only age and sex were considered as covariates for adjustment. The duration of DM is a critical factor that influence the severity of kidney functions. Second, a total of 263 subjects were recruited in this study. The relatively small sample size may limit statistical power. Thus, cohorts from different populations with larger sample sizes are required to confirm our findings. Finally, the correlation between polymorphisms in *UBE3C* and downstream biological pathways should be investigated to understand the molecular mechanisms of disease.

## 5. Conclusions

This study identified novel *UBE3C* variants and haplotypes that are associated with risks of reduced renal functions in Taiwanese DM patients. Further investigation is required to establish the underlying mechanism of the effects from *UBE3C* variants on the pathogenesis of DKD. Our findings provide genetic insights into kidney function-related traits in patients with DKD.

## Figures and Tables

**Figure 1 jpm-10-00210-f001:**
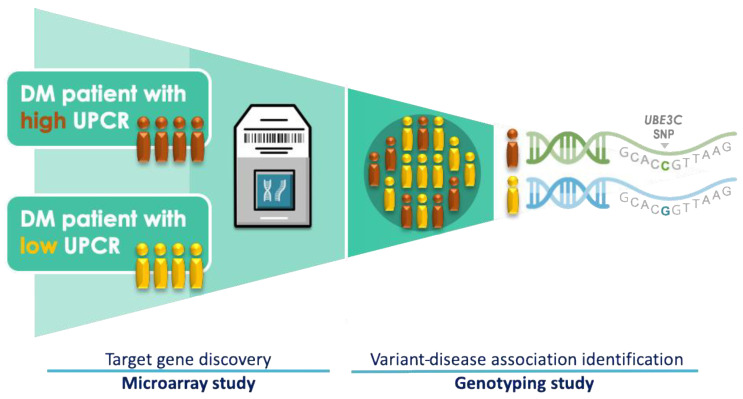
Schematic of the study design. Abbreviations: SNP, single-nucleotide polymorphism; UPCR, urine protein-to-creatinine ratio; DM, diabetes mellitus.

**Figure 2 jpm-10-00210-f002:**
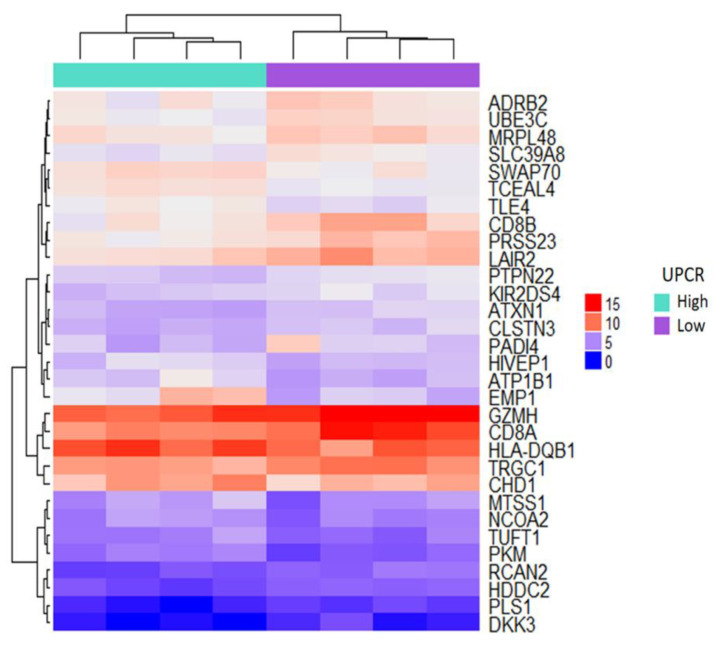
Heat map of significant differentially expressed genes (DEGs) based on the microarray results of patients with high and low expression levels of urine protein-to-creatinine ratio (UPCR). The genes with a *p* value of less than 0.01 and an absolute value of ln (fold change) of more than 0.5 were defined as significant DEGs. An unsupervised, agglomerative hierarchical clustering analysis was employed. Each row represents significant DEGs, and each column represents the patients’ expression profiles.

**Figure 3 jpm-10-00210-f003:**
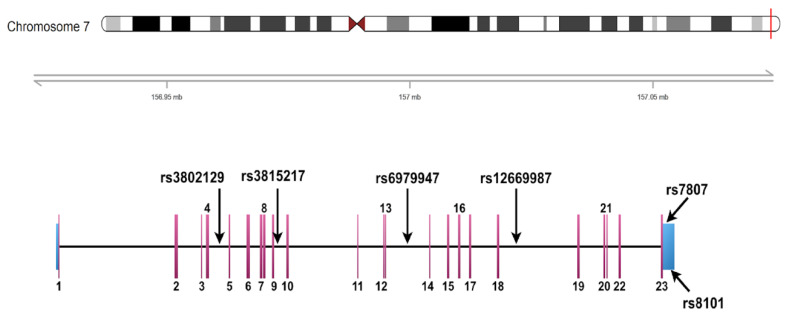
Graphical overview of the genotyped human *UBE3C* gene. Four variants were identified in the intronic region, whereas the other two were identified in the 3-prime untranslated region (3′-UTR).

**Figure 4 jpm-10-00210-f004:**
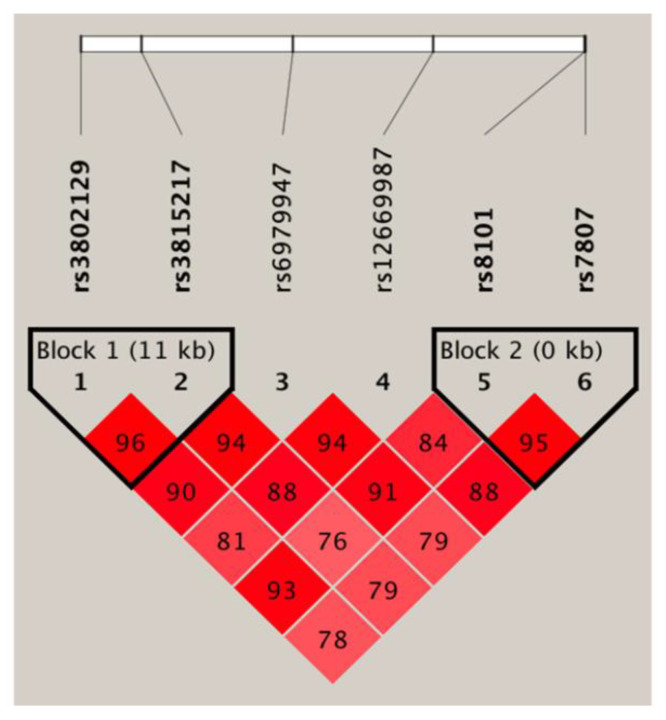
*UBE3C* gene linkage disequilibrium and haplotype block structure in diabetic kidney disease. Based on D-prime, the number on each cell represents D’100.

**Table 1 jpm-10-00210-t001:** Baseline characteristics of the ubiquitin protein ligase E3C (*UBE3C*) single-nucleotide polymorphisms genotyping study population. The case group contained patients with an eGFR <60 mL/min/1.73 m^2^, and the control group contained patients with eGFR ≥60 mL/min/1.73 m^2^.

Characteristics	Case	Control
Total number	172	91
Male, *n* (%)	104 (60.47)	47 (51.65)
Age (years) ^1^	70.47 ± 12.75	65.18 ± 11.89
BMI (kg/m^2^) ^1^	26.94 ± 4.44	26.69 ± 4.69
Smoking, *n* (%)	36 (20.93)	21 (23.08)
HbA1c (%) ^1^	7 ± 1.33	7.17 ± 1.68
eGFR (ml/min/1.73 m^2^) ^1^	32.81 ± 16.11	89.58 ± 24.31
UPCR (mg/g) ^1^	2174.52 ± 3090.89	784.51 ± 1461.02

^1^ Mean ± standard deviation; Abbreviations: BMI, body mass index; HbA1C, hemoglobin A1c; eGFR, estimated glomerular filtration rate; UPCR, urine protein-to-creatinine ratio.

**Table 2 jpm-10-00210-t002:** Basic characteristics of the tag SNPs of *UBE3C*.

				Frequency					
SNP	Position ^a^	Ref	Alt	AFR	AMR	ASN	EUR	TWB	Study Cohort ^b^
rs3802129	chr7:156965297	G	A	0.18	0.44	0.48	0.33	0.44	0.42
rs3815217	chr7:156976981	A	G	0.08	0.44	0.48	0.47	0.50	0.46
rs6979947	chr7:157005863	A	G	0.24	0.37	0.44	0.26	0.40	0.40
rs12669987	chr7:157032676	C	T	0.00	0.05	0.14	0.00	0.15	0.15
rs8101	chr7:157061474	T	C	0.86	0.57	0.52	0.66	0.54	0.52
rs7807	chr7:157061642	C	A	0.71	0.72	0.6	0.79	0.63	0.63

^a^ Reference version: GRCh37. ^b^ Alternative allele frequencies in the *UBE3C* single-nucleotide polymorphisms genotyping study population. Frequencies are shown as alt allele frequency. Abbreviations: AFR, African; AMR, American; ASN, Asian; SNP, single-nucleotide polymorphism; TWB, Taiwan Biobank; Ref, reference allele; Alt, alternative allele.

**Table 3 jpm-10-00210-t003:** Association between *UBE3C* single-nucleotide polymorphisms and chronic kidney disease susceptibility. The case group included patients with an estimated glomerular filtration rate (eGFR) of <60 mL/min/1.73 m^2^, and the control group included patients with an eGFR of ≥60 mL/min/1.73 m^2^.

		Number			Recessive Model	
SNPs	Genotypes	Case (%)	Control (%)	OR (95% CI)	*p* Value	*q* Value
rs3802129	G/G-A/G	127 (75.6)	80 (88.9)	1.00	**0.008**	**0.024**
	A/A	41 (24.4)	10 (11.1)	2.66 (1.24–5.68)		
rs3815217	A/A-A/G	115 (75.2)	74 (87.1)	1.00	**0.032**	0.064
	G/G	38 (24.8)	11 (12.9)	2.19 (1.04–4.62)		
rs6979947	A/A-A/G	128 (77.1)	74 (87.1)	1.00	0.063	0.076
	G/G	38 (22.9)	11 (12.9)	1.97 (0.94–4.15)		
rs12669987	C/C-C/T	149 (96.1)	83 (95.4)	1.00	0.641	0.641
	T/T	6 (3.9)	4 (4.6)	0.73 (0.19–2.74)		
rs8101	C/C-C/T	116 (69.9)	71 (81.6)	1.00	**0.045**	0.068
	T/T	50 (30.1)	16 (18.4)	1.92 (1.00–3.68)		
rs7807	A/A-A/C	136 (80.5)	81 (94.2)	1.00	**0.003**	**0.018**
	C/C	33 (19.5)	5 (5.8)	3.76 (1.39–10.13)		

*p* values were adjusted for age and sex. The *p* and *q* values of <0.05 are displayed in bold; *q* values of <0.05 were considered to be statistically significant after correction for multiple testing.

**Table 4 jpm-10-00210-t004:** Association between haplotype frequencies of the *UBE3C* gene in patients with diabetic kidney disease.

Haplotypes	Frequency		OR (95% CI)	*p* Value
	Case	Control		
*Block 1 (rs3802129/rs3815217)*				
A/G	0.464	0.315	1.76 (1.19–2.61)	**0.005**
G/G	0.051	0.074	0.91 (0.39–2.12)	0.835
A/A	0.004	0.019	0.22 (0.02–2.13)	0.189
G/A	0.482	0.592	Reference	
*Block 2 (rs8101/rs7807)*				
T/C	0.395	0.297	1.58 (1.06–2.35)	**0.024**
C/C	0.010	0.007	1.46 (0.23–9.37)	0.687
T/A	0.121	0.114	1.21 (0.68–2.18)	0.519
C/A	0.474	0.582	Reference	

*p* values of <0.05 are displayed in bold.

**Table 5 jpm-10-00210-t005:** *UBE3C* tag single-nucleotide polymorphism expression–based quantitative trait locus results from the GTEx database.

SNPs	Ref	Alt	Gencode ID (ENSG00000-)	Gene	*p* Value	Effect Size	Tissue
rs3802129	G	A	9335.13	*UBE3C*	0.0000033	0.11	Esophagus-Muscularis
			9335.13	*UBE3C*	0.00001	0.11	Muscle-Skeletal
rs3815217	A	G	229660.1	*RP5–1142J19.1*	3.10 × 10^−9^	−0.37	Skin–Suprapubic ^1^
			229660.1	*RP5–1142J19.1*	0.0000041	−0.29	Skin–Lower leg ^2^
			229660.1	*RP5–1142J19.1*	0.000016	−0.33	Adipose-Visceral
rs8101	T	C	9335.13	*UBE3C*	0.000008	−0.1	Esophagus-Muscularis
rs7807	C	A	9335.13	*UBE3C*	1.40 × 10^−^^8^	−0.15	Esophagus-Muscularis
			9335.13	*UBE3C*	0.000024	−0.15	Artery-Tibial

^1^ No sun exposure; ^2^ sun exposure. Data Source: GTEx Analysis Release V7 (dbGaP Accession phs000424.v7.p2). GTEx, Genotype–Tissue Expression Project.

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
