# Peer review of "Association of UBE3C Variants with Reduced Kidney Function in Patients with Diabetic Kidney Disease"

_jpm, 2020, doi:10.3390/jpm10040210_

Round 1

Reviewer 1 Report

Chen et al present an article entitled "Association of UBE3C Variants With Reduced Kidney Function in Patients with Diabetic Kidney Disease". In this interesting paper, the authors used micorrray and detected interesting variants of UBE3C in patients with DKD. A total of 263 patients were included in the study, comprising 172 patients with DKD and 91 control subjects. Two UBE3C variants were found to be associated with reduced kidney function and associated with higher risks of CKD phenotypes. These findings suggest a clinical role of UBE3C variants in DKD risk

Describe in more details how eGFR was estimated please

Minor

line 151 correct "Ina ddition" please

line 191-92 correct "is similar that indicated the quality of genotyping here" please

line 231 correct "Analysus" please

line 247 correct "indetified" please

Author Response

Point 1: Describe in more details how eGFR was estimated please

Response 1: Thank you for the comments. Estimated GFR (eGFR) was calculated using the four-variable Modification of Diet in Renal Disease equation. Detailed equation is described as following: GFR in mL/min per 1.73 m2 = 175 x SerumCr-1.154 x age-0.203 x 1.212 (if patient is black) x 0.742 (if female).

Ponit 2: Minor

line 151 correct "Ina ddition" please

line 191-92 correct "is similar that indicated the quality of genotyping here" please

line 231 correct "Analysus" please

line 247 correct "indetified" please

Response 2:-> Thank you so much for the valuable comments. We have corrected all the typos.

Reviewer 2 Report

This paper by Chen et al initially appears to be a candidate gene study but in fact is slightly more innovative than that, using a separate analysis to first identify differentially expressed genes before advancing one (UBE3C) for genotyping. I found the analysis to be sound and the study to overall be believable (or at least, as believable as is possible in such a small sample size). My main critique is that the analyses are not as well motivated as they could be, so I would suggest that the introduction be expanded a bit to justify the approach.

Specifically, no hypothesis is stated at the outset. There should be a paragraph that reviews prior data on CKD to suggest the hypothesis that differentially expressed genes could contribute to CKD, and that the differential expression could be in part genetically mediated. This would then make it clearer to understand the design. I had to re-read it several times to understand what the DEG analysis was, since it was presented rather abruptly in the results section.

One somewhat major critique is that the tissues tested in the differential expression and eQTL analyses are not clearly described. This is obviously quite important, since the DEG analysis is used to drive the whole study. In addition, it is not clear whether the same tissues are used for the DEG analysis, the GTEx eQTL analysis, or the eQTL analysis in the current samples.

I felt that the DEG results were rather qualitative and it was hard to understand exactly what was done, or how significant they were. More quantitative results would be helpful.

Why was the recessive model tested? Was the additive model tested? Either the recessive test should be motivated, or if other models were tested the p-values should be corrected for all analyses

The variants identified are stated as being GTEx eQTLs. Are these eQTLs co-incident with the genetic associations? A co-localization analysis would ideally be performed (although this is not a major issue since the eQTLs are not the major storyline here)

The haplotype analysis was also not motivated. In addition, the model tested here seems to be additive (unless I misunderstand). At any rate, the testing of a recessive model initially, and then a haplotype model here, need to be justified and connected.

Author Response

Please see the file attached. Thank you. 

Reviewer 3 Report

Chen and coworkers examined UBE3C gene variants association with reduce renal function in patients with diabetic kidney disease. The manuscript should be reviewed for Grammer/Syntax errors.

Author Response

Chen and coworkers examined UBE3C gene variants association with reduce renal function in patients with diabetic kidney disease. The manuscript should be reviewed for Grammer/Syntax errors.

Response: Many thanks your comments and suggestions. Grammar and writing style errors have been corrected by our colleague who is a native English speaker.